# Is socioeconomic segregation of the poor associated with higher premature mortality under the age of 60? A cross-sectional analysis of survey data in major Indian cities

Tarani Chandola,[1] Sitamma Mikkilineni,[2] Anil Chandran,[3] Souvik Kumar Bandyopadhyay,[4] Nan Zhang,[1] Sergio Luiz Bassanesi[5]

[1]Department of Social Statistics, The Cathie Marsh Institute, University of Manchester, Manchester, UK
[2]IBS Business School, IFHE University, Hyderabad, India
[3]Department of Demography and Population Studies, University of Kerala, Thiruvananthapuram, India
[4]Indian Institute of Public Health, Public Health Foundation of India, Hyderabad, India
[5]Departamento de Medicina Social, Universidade Federal do Rio Grande do Sul, Porto Alegre, Brazil

**Correspondence to**
Dr Tarani Chandola;
tarani.chandola@manchester.ac.uk

## ABSTRACT

**Objectives** Although urbanisation is generally associated with poverty reduction in low-income and middle-income countries, it also results in increased socioeconomic segregation of the poor. Cities with higher levels of socioeconomic segregation tend to have higher mortality rates, although the evidence is based on ecological associations. The paper examines whether socioeconomic segregation of the poor is associated with higher under-60 years ('premature') mortality risk in Indian cities and whether this association is confounded by contextual and compositional sociodemographic and socioeconomic factors.

**Setting and participants** A population representative sample of over one million from 39 427 households living in 1876 urban wards within 59 Indian districts (cities) from the third (2008) District Level Household Survey (DLHS-3).

**Primary outcome and other measures** The outcome was any death under the age of 60 reported by households in the preceding 4 years of the DLHS-3. Socioeconomic segregation, estimated at the district (city) level, was measured using an isolation index of the poor and the index of dissimilarity.

**Results** Poor households living in cities where the poor were more isolated had higher probabilities of premature mortality than poor households living in cities where the poor were less isolated. In contrast, it did not matter whether rich households lived in more or less socioeconomically segregated cities. A 1 SD increase in the isolation index was associated with an absolute increase of 1.1% in the probability of premature mortality for the poorest households.

**Conclusion** Increasing segregation of the poor may result in higher premature mortality. As low-income and middle-income countries become increasingly urbanised, there is a risk that this may lead to increased segregation of the poor as well as increased premature mortality.

## Strengths and limitations of this study

► The study examines how residential socioeconomic segregation is related to premature mortality.
► This is an important topic because urbanisation in low-income and middle-income countries is accompanied by an increasing spatial concentration of poverty in cities.
► For poor households, the increased risk of premature mortality associated with urban socioeconomic segregation remains even after taking into account household and city level socioeconomic and sociodemographic confounders.
► This is a cross-sectional study so we were unable to account for unobserved factors that could cause the association between socioeconomic segregation and higher premature mortality.
► We have not estimated any lagged effects of socioeconomic segregation on premature mortality.

Although urbanisation can play a positive role in overall poverty reduction, an increase in the socioeconomic segregation of poor people living in cities may have adverse outcomes in terms of their chances for good health and life opportunities. While levels of urbanisation in India are relatively low at around 31% in 2011, the process of urbanisation is among the fastest in the world. As a result of urbanisation, India contains the highest number of urban slum dwellers, accounting for 17% of the world's slum dwellers.[1]

The association between poverty and ill health is well documented. The risk of premature mortality and disease as a result of living in poor areas such as slums with inadequate housing, lack of access to safe water and adequate sanitation is well known. Furthermore, there are now a multitude of studies showing associations between

## INTRODUCTION

Urbanisation in low-income and middle-income countries is accompanied by increasing spatial concentration of poverty in cities.

area level poverty rates and deprivation and poor health/mortality, even after taking into account individual and household level socioeconomic circumstances.[2–4] This suggests an area or contextual effect on health associated with living in a poor area, which is over and above the compositional effect of the number of poor people in an area. What is perhaps less well known is the association between socioeconomic segregation and health. Studies have shown that there are ecological correlations between socioeconomic segregation by income and mortality rates.[5–8] These studies have shown that cities where there is greater socioeconomic segregation of the poor tend to have higher mortality rates. Furthermore, it is also likely that poor people living in more socioeconomically segregated cities have poorer health than other poor people who are living in less segregated cities. The concept of a 'triple health jeopardy'[5] of being poor, living in a poor district that is socioeconomically segregated from the rest of the city has been suggested. But there has been little empirical evidence linking individual mortality records to household poverty and area level segregation.

There may be a number of mechanisms underlying the relationship between socioeconomic segregation and mortality. Poor nutrition, substandard housing conditions and overcrowding increase susceptibility and exposure to infectious diseases.[9–12] Poor people living in socioeconomically segregated cities may have less access to good quality public services such as roads and transport infrastructure and restricted access to primary care services[5 7 13] compared with poor people living in less segregated cities. Poor neighbourhoods are more likely to be exposed to environmental pollutants,[14–17] and their inhabitants may consequently have higher risks of respiratory and cardiovascular diseases and cancers.[18 19]

However, there are few, if any analyses that take into account potential confounders of the ecological correlation between cities with higher levels of socioeconomic segregation and mortality rates. An apparent contextual effect of living in socioeconomically segregated cities on mortality may actually be a reflection of individual and household level associations between socioeconomic circumstances and health. Furthermore, the association between socioeconomic segregation and mortality may also be confounded by other contextual level factors such as poverty rates in the city. As people from poor households tend to live in poorer neighbourhoods and are also more likely to live in poorer cities, any association between socioeconomic segregation and health needs to take account of such potential confounding from socioeconomic factors at the compositional (household or individual) as well as contextual (city) levels.

The main research question of this paper is to investigate whether being poor, living in a poor area that is socioeconomically isolated from the rest of the city is associated with greater mortality risk compared with poor people living in less socioeconomically segregated areas in India.

## METHODS
### Datasets
Data from the third District Level Household Survey (DLHS-3)[20] were analysed, with the survey conducted between 2007 and 2008. The DLHS was designed to provide information on reproductive and child health in all the districts of India. The steps involved in the sampling of households were:

i. the selection of primary sampling units (PSUs) within census wards;
ii. the selection of the households from each of the selected village/urban PSUs through a random selection of households within a census enumeration block (which we have used to define as a neighbourhood in this study).

The overall household response rate, the number of households interviewed per targeted 100 households, was 94%. The DLHS-3 PSUs were made available for secondary data analyses, allowing the estimating of neighbourhood socioeconomic segregation indices.

For this study, the analytical sample was restricted to households living within urban wards of the largest cities with a population of more than one million. This resulted in an analytical sample of 39 446 households living in 1876 urban wards within 59 cities. This was to ensure comparability of the urban units within the analysis. Mega cities such as Kolkata, Mumbai and Delhi were disaggregated into their constituent administrative districts.

### Outcome
Premature mortality under the age 60 was reported for each household participating in the DLHS-3 in the preceding 4 years (from 2004 to 2008). This was recoded into a binary variable (any premature death in the household vs no premature death) for the probit regression analysis. The under-60 death rate was calculated at the district level by dividing the total premature mortality in a district by the number of respondents aged under 60 in the DLHS-3 sample.

### Measures of socioeconomic segregation
There are a number of measures for segregation, measuring different dimensions of the concept. In this study, two measures of segregation were used—the isolation index of segregation and the index of dissimilarity.

#### The isolation index of segregation
The study used estimates of neighbourhood socioeconomic composition from DLHS-3 districts and PSUs. PSUs correspond to around 300 households from a census enumeration block and we used them to represent urban 'neighbourhoods'. Poverty was measured through the proxy indicator of illiteracy as direct measures of poverty were not available in the data. The index of isolation was used to measure the socioeconomic segregation of the poor. The isolation index ($Ind\_isolation_j$) measures the extent to which poor members of a

district are exposed only to one another and is computed as the weighted average of the proportion of poor people in each neighbourhood.[21]

$$\text{Ind\_isolation}_j = \sum_{i=1}^{j} \frac{\text{illiterate}_{ij}}{\text{illiterate}_j} \times \frac{\text{illiterate}_{ij}}{\text{population}_{ij}}$$

where $\text{illiterate}_{ij}$ is the number of illiterate adults living in $\text{neighbourhood}_i$ within $\text{district}_j$, $\text{illiterate}_j$ is the number of illiterate adults living in $\text{district}_j$ and $\text{population}_{ij}$ is the adult population living in $\text{neighbourhood}_i$ within $\text{district}_j$.

As the isolation index is a weighted poverty rate of the district, it is strongly influenced by the (unweighted) poverty rate in a district. It can be adjusted to control for the effect of population composition in the city by the following way:

$$\text{Ind isolation adj}_j = (\text{Ind isolation}_j - P)/(1 - P)$$

where P is the proportion of poor people in the city.

This adjusted isolation index has also been suggested as a potential measure of segregation in its own right. Stearns and Logan[22] go so far as to suggest that it represents an independent dimension of segregation.

### Index of dissimilarity

Another measure of segregation in this study is the index of dissimilarity which focuses on residential evenness.[23] It measures the evenness with which poor and rich groups are distributed across the neighbourhoods that make up a city.

The index indicates the proportion of the poor population that would have to move neighbourhoods in order to achieve an even distribution within the city.

$$\text{Ind\_dissimiliarilty}_j = 0.5 \left| \sum_{i=1}^{j} \frac{\text{illiterate}_{ij}}{\text{illiterate}_j} - \frac{\text{not illiterate}_{ij}}{\text{not illiterate}_j} \right|$$

For all the segregation measures, the DLHS-3 weights were used to derive the district and neighbourhood level estimates of the illiterate, not illiterate and total populations. As the DLHS-3 does not contain information on all neighbourhoods within a district, the estimated indices may not accurately reflect the true distribution of the district level segregation indices. However, the use of the DLHS-3 data does allow for an estimate of segregation using data from smaller geographical areas (through the random sample of households within a PSU), rather than estimating the index using data from larger census area levels such as the ward level, which often refers to a very large area and population (around 25 000–75 000) for the largest Indian cities.

### Covariates

As there are no direct measures of income or poverty in the DLHS data, poverty was measured through the use of a proxy indicator namely illiteracy. Another reason for using illiteracy to measure poverty is the difference in the meanings and concepts of standard poverty measures (such as indicators of wealth and status) across Indian

cities. The illiteracy rates were calculated by dividing the total number of illiterate adults in a DLHS-3 household or district by the number of adults in that household or district. Other potential sociodemographic and socioeconomic confounders included the number of household members, household wealth (constructed by the DLHS-3 team from a factor analysis of consumption measures and housing quality indicators) and religion of head of household, whether the head of household belonged to a scheduled caste (SC) or scheduled tribe (ST), and whether the household accessed government or private healthcare when someone was ill. City population (per 100 000 inhabitants) and proportion of city population that belonged to SC or ST groups were derived from the weighted DLHS-3 district level estimates.

### Analysis

The ecological (city level) correlation between the district level under-60 mortality rate, the poverty rate and the three socioeconomic segregation measures was examined. Multilevel probit regression (with the outcome of under-60 death within a household) was used to estimate the effect of the (district level) three measures of socioeconomic segregation, after taking into account socioeconomic, sociodemographic factors at the individual and household levels, as well as city level socioeconomic factors (the poverty rate, the population and proportion of population that are SC/ST). Multilevel analysis is particularly appropriate as it takes into account the sample design of the DLHS-3 which results in clustered samples at the state, district and neighbourhood (PSU) levels—each of these levels was specified with a random intercept in the multilevel models. It is important to take this clustering into account in the analysis as the main exposure variable is at the district level, rather than at the household level. The multilevel regression coefficients were estimated using the penalised quasi-likelihood (PQL2) estimation procedures within MLwiN.

The question of whether being poor, living in a poor area that is socioeconomically segregated from the rest of the city is associated with greater mortality risk was examined through an interaction between the household illiteracy rate ('being poor') with the measures of socioeconomic segregation.

### RESULTS

The scatterplot of the district level associations between premature death rates and the unadjusted Index of Isolation (figure 1) shows that there was a moderate ecological correlation (r=0.52). Districts (cities) where the poor were more isolated were also those with higher premature death rates. Only a few of the 59 districts are displayed in the scatterplot to aid ease of interpretation. Some of the districts such as North West Delhi and West Delhi had the same levels of the isolation index, but differed in terms of premature death rates by more than 5%. The ecological correlations (for all the 59 districts) between

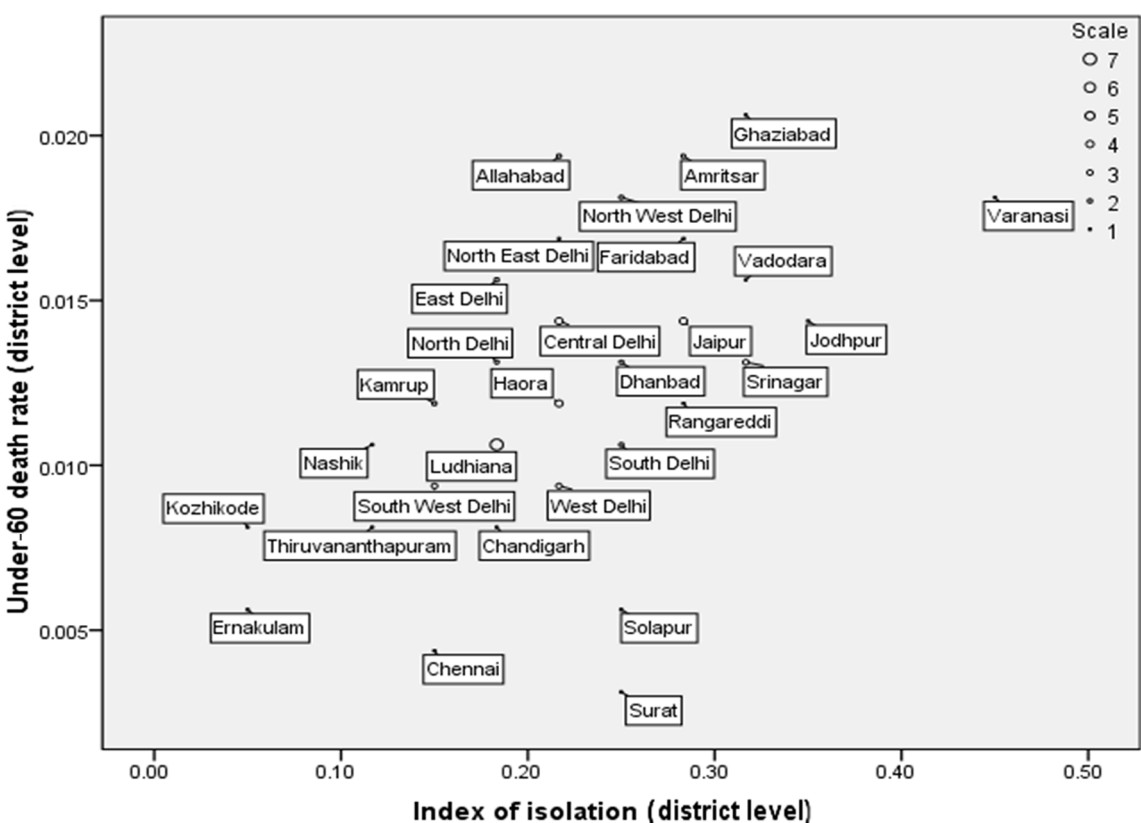

**Figure 1** Scatterplot of district level under-60 death rate and the unadjusted index of Isolation in 59 Indian urban districts—third District Level Household Survey (DLHS-3) (2008): selected districts shown.

all the district level measures are shown in table 1. The correlation between the (unadjusted) index of isolation and illiteracy rate was very strong (r=0.90) which meant that using both in a regression analysis would be likely to cause multicollinearity problems. In contrast, there was a weak correlation between the adjusted index of isolation and the illiteracy rate, and a strong correlation between the adjusted index of isolation and the dissimilarity index.

Table 2 displays the distribution of the key independent variables in the analysis by any premature death in the household. 5.3% of all households in the analytical sample reported the death of a household member aged under 60 in the 4-year period preceding the survey. Households where there was a premature death in the preceding 4 years were living in districts with higher index of isolation scores (unadjusted and adjusted indices), lower dissimilarity index scores, higher illiteracy rates at the

district and household levels, lower wealth scores, more household members, were more likely to be Muslim and SC/ST households and also were more likely to use government clinics when ill.

Table 3 displays the results (estimated probits and 95% CIs) of the multilevel probit regression models with any premature death in the household as the dependent variable. In model 1, the model with all the household level sociodemographic and socioeconomic factors was fitted, taking into account the clustering in premature mortality at the district, PSU and household levels. This probit model allows for extra-binomial variance at the lowest (household) level as the variance of the distribution of household mortality was greater than expected under the binomial assumption. There appeared to be some clustering in premature mortality at the PSU level, but a negligible amount of clustering at the district level.

**Table 1** Ecological correlation between district level under-60 death rate, illiteracy rate and measures of socioeconomic segregation

| | Under-60 death rate | Illiteracy rate | Index of isolation (unadjusted) | Index of isolation (adjusted) | Index of dissimilarity |
|---|---|---|---|---|---|
| Under-60 death rate | 1 | | | | |
| Illiteracy/poverty rate | 0.50 | 1 | | | |
| Index of isolation (unadjusted) | 0.52 | 0.90 | 1 | | |
| Index of isolation (adjusted) | 0.30 | 0.29 | 0.68 | 1 | |
| Index of dissimilarity | −0.15 | −0.34 | 0.04 | 0.67 | 1 |

**Table 2** Distribution (mean and percentages) of key independent variables in the analysis by premature death (under 60) in the household: DLHS-3

| | No death under 60 in the household | ≥1 death under 60 in the household |
|---|---|---|
| n | 37 346 | 2100 |
| Isolation index (district level) | 0.22 (0.06) | 0.23 (0.06) |
| Isolation index adjusted (district level) | 0.08 (0.03) | 0.09 (0.04) |
| Dissimilarity i ndex (d istrict level) | 0.33 (0.07) | 0.32 (0.07) |
| Illiteracy rate (d istrict level) | 0.15 (0.05) | 0.16 (0.05) |
| Illiteracy rate (household level) | 0.16 (0.24) | 0.23 (0.29) |
| Mean wealth index score | 1.33 (0.93) | 1.05 (0.92) |
| Mean number of household members | 4.92 (2.37) | 5.25 (2.78) |
| % of households that are Muslim | 16.4% | 19.3% |
| % of households that are SC/ST | 18.4% | 24.1% |
| % of households using government clinics when ill | 38.4% | 41.9% |

DLHS-3, third District Level Household Survey; SC, scheduled caste; ST, scheduled tribe.

We additionally examined whether adding in Indian states as an additional level was needed in the multilevel analyses, but found little evidence of clustering in premature mortality at the state level.

Being a member of an illiterate or poorer household with more household members, and living in the Northern region were associated with increased probabilities of premature mortality. Furthermore, living in a city with a higher proportion of households belonging to SCs and STs was associated with higher probabilities of premature mortality, although the population size of cities did not predict mortality. While the main effect of the district level poverty rate on premature mortality was positive and significant (analysis not shown), the interaction term between district level poverty rate and the household level poverty rate was non-significant. This suggests that while there are additive but not multiplicative effects of living in a poor (illiterate) household and poor (illiterate) city on the risks of premature mortality.

Model 2 added in the unadjusted isolation index (in SD unit differences from the mean) and the interaction between the (unadjusted) isolation index and the household illiteracy rate as independent variables. A 1 SD increase in the isolation index was associated with a non-significant increase of 0.02 in the probit of premature mortality for the richest (non-illiterate) households. Whereas for the poorest (all illiterate) households, an increase in 1 SD of the (unadjusted) isolation index was associated with a significant increase of 0.15 in the probit of premature mortality (0.02+0.13). As the unadjusted isolation index and the district level illiteracy rate are strongly correlated, these probit estimates may suffer from collinearity issues. Hence we used the adjusted isolation index in model 3 to avoid potential collinearity issues with the district level illiteracy rate. Here again, there was some evidence of an interaction between the (adjusted) isolation index and household level illiteracy rate—poor households (where all the members were illiterate) living in more isolated cities (a 1 SD increase in the adjusted isolation index) had significantly higher probits of premature mortality (0.10) compared with poor households living in cities where the poor were less isolated. The estimated probabilities from model 3 are graphed in figure 2. Although richer (completely literate) households have lower probabilities of premature mortality than poorer (completely illiterate) households, the poorest households which live in cities where the poor are less isolated have significantly lower probabilities of premature mortality than the poorest households living in cities where the poor are more isolated. The estimated probability of premature mortality for the poorest households living in cities where the adjusted isolation index was 1 SD below the average for all the Indian cities was 5.5%. At average levels of of the adjusted isolation index, the probability of premature mortality for the poorest households was 6.6%—an absolute difference of 1.1%. In contrast, for rich households, there was hardly any difference in the predicted probability of premature mortality in terms of living in city where the poor were more or less isolated. A very similar interaction effect was found in model 5 for the combination of poor households living in cities with a greater dissimilarity index, but the interaction term was borderline significant.

## DISCUSSION

This study found that urban Indian districts where the poor were segregated (ie, isolated or unevenly distributed) had higher premature mortality rates than districts where the poor were less segregated. Poor households living in cities where the poor were less isolated (cities which were 1 SD lower than the average adjusted isolation index of Indian cities) had around 1.1% lower probabilities of premature mortality than poor households living

**Table 3** Probits (and 95% CIs) from multilevel probit regression models of premature mortality: DLHS-3

| | Model 1 | Model 2 | Model 3 | Model 4 |
|---|---|---|---|---|
| *Fixed part* | | | | |
| Intercept | −1.60 (−1.67 to −1.53) | −1.59 (−1.66 to −1.52) | −1.59 (− 1.66 to −1.52) | −1.59 (−1.66 to −1.52) |
| Household illiteracy rate (ref: none illiterate) | | | | |
| Some members are illiterate | 0.10 (0.04 to 0.15) | 0.10 (0.04 to 0.15) | 0.09 (0.04 to 0.15) | 0.10 (0.05 to 0.15) |
| All members are illiterate | 0.17 (0.05 to 0.29) | 0.16 (0.04 to 0.28) | 0.19 (0.08 to 0.30) | 0.22 (0.11 to 0.32) |
| Wealth tertiles (ref: poorest tertile) | | | | |
| Middle wealth tertile | −0.11 (−0.16 to −0.06) | −0.11 (−0.16 to −0.06) | −0.11 (−0.16 to −0.06) | −0.11 (−0.16 to −0.06) |
| Richest tertile | −0.25 (−0.32 to −0.19) | −0.25 (−0.32 to −0.19) | −0.25 (−0.31 to −0.19) | −0.25 (−0.31 to −0.19) |
| Region (ref: North) | | | | |
| East | −0.15 (−0.23 to −0.06) | −0.15 (−0.24 to −0.07) | −0.15 (−0.23 to −0.06) | −0.15 (−0.24 to −0.07) |
| West | −0.11 (−0.19 to −0.03) | −0.11 (−0.19 to −0.03) | −0.11 (−0.19 to −0.03) | −0.11 (−0.19 to −0.03) |
| South | −0.23 (−0.31 to −0.15) | −0.23 (−0.31 to −0.14) | −0.23 (−0.31 to −0.14) | −0.23 (−0.31 to −0.15) |
| Religion (ref: Hindu) | | | | |
| Muslim | 0.02 (−0.04 to 0.08) | 0.02 (−0.04 to 0.09) | 0.02 (−0.04 to 0.09) | 0.02 (−0.04 to 0.09) |
| Other religion | 0.03 (−0.06 to 0.12) | 0.03 (−0.06 to 0.12) | 0.03 (−0.06 to 0.12) | 0.03 (−0.06 to 0.12) |
| Caste (ref: other caste hh) | | | | |
| SC/ST caste hh | 0.11 (0.05 to 0.17) | 0.11 (0.05 to 0.17) | 0.11 (0.05 to 0.17) | 0.11 (0.05 to 0.17) |
| Other backward caste hh | 0.05 (−0.01 to 0.10) | 0.04 (−0.01 to 0.10) | 0.04 (−0.01 to 0.10) | 0.05 (−0.01 to 0.10) |
| Missing/DK caste hh | −0.05 (−0.22 to 0.13) | −0.05 (−0.23 to 0.13) | −0.05 (−0.23 to 0.12) | −0.05 (−0.23 to 0.13) |
| Access to private healthcare (ref: yes) | | | | |
| Usually go to government clinic when ill | 0.01 (−0.03 to 0.06) | 0.01 (−0.03 to 0.06) | 0.01 (−0.03 to 0.06) | 0.01 (−0.03 to 0.06) |
| Number of people in the hh | 0.01 (0.003 to 0.02) | 0.01 (0.003 to 0.02) | 0.01 (0.003 to 0.02) | 0.01 (0.003 to 0.02) |
| % of pop who are SC/ST (district level) | 0.004 (0.0003 to 0.01) | 0.004 (0.0001 to 0.01) | 0.004 (0.0001 to 0.01) | 0.004 (0.00003 to 0.008) |
| Population per 100 000 (district level) | 0.0001 (−0.002 to 0.002) | 0.0001 (−0.002 to 0.002) | 0.00003 (−0.002 to 0.002) | 0.00004 (−0.002 to 0.002) |
| Illiteracy rate (district level) | 0.63 (−0.07 to 1.34) | 0.24 (−0.81 to 1.29) | 0.73 (0.17 to 1.29) | 0.84 (0.28 to 1.41) |
| Interaction of illiteracy rate (district level) and illiteracy rate (hh) | | | | |
| Illiteracy rate*Some illiterate | 0.14 (−0.77 to 1.06) | | | |
| Illiteracy rate*All illiterate | 1.89 (−0.24 to 4.01) | | | |
| Unadjusted isolation index (district level) | | 0.02 (−0.04 to 0.07) | | |
| Interaction of unadjusted isolation index and (district level) and illiteracy rate (hh) | | | | |

Continued

**Table 3** Continued

| | Model 1 | Model 2 | Model 3 | Model 4 |
|---|---|---|---|---|
| Unadjusted isolation index*Some illiterate | | 0.02 (−0.03 to 0.07) | | |
| Unadjusted isolation index*All illiterate | | **0.13 (0.02 to 0.23)** | | |
| Adjusted isolation index corrected (district level) | | | −0.004 (−0.04 to 0.03) | |
| Interaction of adjusted isolation index adjusted and (district level) and illiteracy rate (hh) | | | | |
| Adjusted isolation index*Some illiterate | | | 0.03 (−0.01 to 0.07) | |
| Adjusted isolation index*All illiterate | | | **0.10 (0.01 to 0.19)** | |
| Dissimilarity index (district level) | | | | −0.01 (−0.04 to 0.03) |
| Interaction of dissimilarity index and (district level) and illiteracy rate (hh) | | | | |
| Dissimilarity index*Some illiterate | | | | 0.03 (−0.02 to 0.07) |
| Dissimilarity index*All illiterate | | | | 0.11 (−0.0005 to 0.22) |
| Random part | | | | |
| District variance (n=59) | 0.013 (0.014) | 0.013 (0.014) | 0.013 (0.014) | 0.013 (0.014) |
| PSU variance (n=1876) | 0.038 (0.017) | 0.037 (0.017) | 0.036 (0.017) | 0.036 (0.017) |
| Household extra binomial variance (n=39 427) | 0.934 (0.007) | 0.934 (0.007) | 0.935 (0.007) | 0.936 (0.007) |

Statistically significant (P<0.05) probit coefficients are shown in bold.
DK, don't know; DLHS-3, third District Level Household Survey; hh, household; PSU, primary sampling unit; SC, scheduled caste; ST, scheduled tribe.

in cities where the poor were more isolated. This association appeared to remain even after taking into account a range of compositional (household level) and contextual (district level) sociodemographic and socioeconomic factors that could confound this association. There thus appears to be some evidence of a 'triple health jeopardy'[5] of being poor, living in a poor district that is socioeconomically isolated from the rest of the city.

In India, the top causes of mortality include non-communicable causes like cardiovascular diseases, cancer and respiratory diseases, as well as infectious diseases like tuberculosis, malaria and diarrhoea, and injuries (suicide and accidents).[24] Poor people living in cities where the poor are less isolated may be less exposed to environmental pollutants that cause respiratory diseases and cancers and they may benefit from the shared living spaces with richer people. Conversely, poor people living in cities where the poor are more isolated may be far from good transport and green spaces, and have little access to water, sewage, sanitation and good quality health services in the neighbourhoods where they live. Even though they live in the same city, because the neighbourhoods they live in are very isolated from rich people, so they cannot access the same services as the rich people do. This association between socioeconomic segregation and premature mortality is independent of how poor the city is as a whole, and thus the association appears to be related to the residential distribution of poor people in a city, rather than how rich or poor the city is. Socioeconomic segregation may enable the affluent access to the greatest resources and reinforce the social exclusion of the disadvantaged.[21 25 26] Income segregation may reinforce social inequality[27] through segregating the poor away from resources such as employment, transport, education, healthcare and other services, a process which could discourage social mobility. It may thus capture dimensions of social inequality distinct from poverty or deprivation rates that have typically been used to measure social inequality at an area level.

There are a number of limitations to this study. This is a cross-sectional study showing associations between premature mortality and a single measure of socioeconomic segregation. We were unable to infer causality from observational evidence as we have not observed whether increasing socioeconomic segregation results in higher premature mortality. While we have adjusted for a range of potential socioeconomic confounders at the district and household levels, there may be other unobserved factors, including selection effects that cause the association of socioeconomic segregation with premature mortality. In addition, we have not estimated any lagged effects of socioeconomic segregation on premature mortality. If the association is causal, we do not know what duration of exposure to a socioeconomically segregated neighbourhood is needed to generate higher premature mortality risks. The measure of socioeconomic segregation was estimated from survey data and so may not be accurately estimated. Online supplementary appendix 1 examines the association of the index of isolation with the district level illiteracy rates for the Delhi districts, where there have been some studies on the spatial

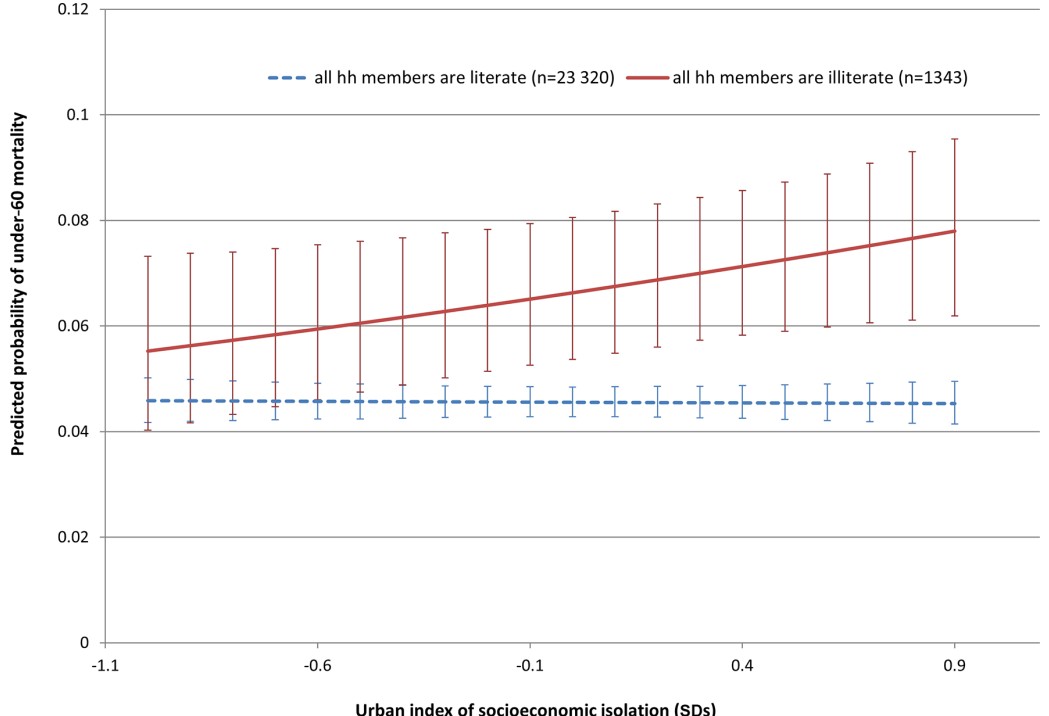

**Figure 2** Predicted probabilities (and 95% CIs) of under-60 mortality from DLHS-3 by the adjusted index of socioeconomic isolation and household literacy: probabilities estimated from model 3, table 3. DLHS-3, third District Level Household Survey; hh, household.

distribution of slum housing. The analysis confirms the expected ranking of Delhi districts by the index of isolation, suggesting that the isolation index estimated by the DLHS may be valid. Other dimensions of socio-economic segregation could not be measured due to data limitations. In particular, we have not been able to measure the spatial dimension of socioeconomic segregation which may be particularly relevant in relation to discussions of social inequalities. The outcome measure, under-60 mortality, was not age-specific, resulting in different causes of deaths being grouped together in a crude metric. This was because of the small numbers of deaths at the household level. If we had attempted more age-specific mortality analyses, this would have resulted in unstable models with zero cell counts.

As urbanisation increases in India, there is a risk of increasing socioeconomic segregation of the poor and associated health risks. Indian cities already contain the largest concentration of the world's population living in slums. With increasing urbanisation and associated economic pressures, the poor living within Indian cities are at risk of increasing isolation such as through slum resettlement programmes far from employment, transport and public services networks. The results from this study cautions against such policies that increase the socioeconomic segregation of the poor, as this may lead to increased levels of premature mortality among the most vulnerable groups in society.

**Contributors**  TC, SKB and SM contributed to the conception and design of the study. ACS, SLB and NZ contributed to the analysis and interpretation of the data. TC drafted the initial manuscript and SLB, SM, SKB, ACS and NZ contributed to revisions of the manuscript. All authors gave approval for the final version and agree to be accountable for all aspects of the study.

**Funding**  This work was supported by the Economic and Social Research Council (ESRC): RES-238-25-0015 and ES/J019119/1.

**Competing interests**  None declared.

**Patient consent**  Not required.

**Ethics approval**  This study was approved by the institutional ethics committee of the International Institute for Population Sciences.

**Provenance and peer review**  Not commissioned; externally peer reviewed.

**Data sharing statement**  Data from the third District Level Household Survey are made available on request through the IIPS website (http://rchiips.org/obtainingdata.html). There are no additional unpublished data used in this manuscript.

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
