## [Reviewer comments · BMJ Open]

ARTICLE DETAILS

TITLE (PROVISIONAL)	Is socioeconomic segregation of the poor is associated with higher premature mortality under the age of 60? A cross-sectional analysis of survey data in major Indian cities
AUTHORS	Chandola, Tarani; Mikkilineni, Sitamma; Chandran S, Anil; Bandyopadhyay, Souvik; Zhang, Nan; Bassanesi, Sergio

VERSION 1 – REVIEW

REVIEWER	Nick Bailey University of Glasgow, UK
REVIEW RETURNED	31-Aug-2017

GENERAL COMMENTS	Socioeconomic segregation of the poor is associated with higher premature mortality under the age of 60 in major Indian cities – Chandola et al The paper uses survey data on household resources and experience of mortality across a number of Indian cities to try to examine whether spatial segregation has an impact on mortality. While the topic is clearly relevant to the journal and the data appear to be of reasonable quality, I believe the design has a serious problem. I also felt that there were weaknesses in framing of the work, in the theory or motivation, and in the discussion or interpretation. All could be remedied but I think the paper cannot be published at this stage. I think there is a basic methodological problem with the paper. The Index of Isolation is computed for districts (cities). It has a maximum of 1 (where the poor are only ever found in neighbourhoods with 100% poverty) but it has a minimum which is equal to the proportion poor in the district (city) as a whole. This occurs where the poor are dispersed in equal concentrations in all neighbourhoods. It cannot go lower than this and the authors are wrong therefore to state that it has a minimum value of zero (page 6, line 27). Cities with more poverty will therefore tend to have a higher Index value without there necessarily being any more segregation. That is why the correlation of these two at District level is so strong (0.9 – page 7, line 39). The implication is that the correlations of district-level Index of Isolation and district-level premature mortality (Figure 1) are at least in part simply correlations of poverty (illiteracy) and mortality. Likewise, the modelling results (Table 2) which suggest a relationship between Isolation and mortality will be at least in part simply showing the impacts of poverty.
--

There is, however, a simple correction factor which can be applied here which removes the effect of composition, as Massey and Denton's (1988) paper noted, citing work by Bell (1954). This ought to produce a much greater separation of district poverty and segregation (although I would expect some correlation to remain), and hence permit both to be controlled in the same models.

At present the two are modelled separately (compare models 2 and 4 in Table 2). The effects are near-identical overall and when treated as interactions with household poverty (models 3 and 5). On the latter, the paper claims there is a major difference here presumably focussing on the significance tests. The estimated regression coefficients are very similar.

In the results summary (line 25) and in the findings (page 8, line 40), I think the paper gives an exaggerated impression of the scale of the effects observed. The scale is described in relation to a unit increase in the isolation index. As that is beyond the limits of possible change (see above), it is not an appropriate basis for reporting. The observed range is less than 0.5 (Figure 1). A better basis might be change in mortality in relation to a one standard deviation change in the Index of Isolation for the set of districts, perhaps. That would of course make the reported effect correspondingly smaller – somewhere around 2% change in mortality.

The paper risks over-generalising. Segregation is a complex, multi-dimensional concept as Massey and Denton's (1988) paper shows. In this paper, the focus is on the dimensions of 'exposure' or rather its converse, isolation. It might be better to refer to isolation throughout the analytical stage. This is made more important, perhaps, by the use of illiteracy as the measure of poverty so there is a twofold move – from the illiteracy/isolation relationship to poverty/segregation.

The paper uses the term segregation too loosely in another sense. Segregation is an attribute of 'cities' or systems of neighbourhoods. Individual neighbourhoods cannot be 'segregated' – they are only poorer or richer. A poor household in a poor neighbourhood can be more isolated but not segregated. There are numerous examples where the paper uses 'segregation' as a term related to the neighbourhood.

The first sentence of the abstract suggests that increased socio-economic segregation might not occur along with urbanisation when it is hard to see how it would not. The same point is made at start of the introduction (p4, line 3). Any segregation in rural communities would be very much more limited so urbanisation must imply a shift to more segregated contexts. In any case, the paper does not compare urban and rural but explores variations between cities so this statement is not needed.

The paper argues that spatial inequalities in health persist even after controlling for individual/household factors (page 4, line 22) and argues that this is evidence for contextual effects on health. This needs to be much more cautiously stated. These observations can never fully account for individual/household factors so cannot be taken as strong evidence for contextual effects, not least because there is such an obvious risk of selection bias: the 'poor' living in poorer neighbourhoods are likely to be poorer than those in better neighbourhoods in ways we have not observed.

	These selection effects are central to the neighbourhood effects literature. Other minor points The outcome measure is a very crude mortality rate which ignores age composition. For the survey, no information is provided on response rates which give some indication of data quality. Table 2 – reference category for household illiteracy is presumably 'none illiterate'.
--	---

REVIEWER	Trevon Logan The Ohio State University
REVIEW RETURNED	05-Sep-2017

GENERAL COMMENTS	The authors should also use the dissimilarity index in addition to isolation to measure the effects. Although the two measures are different they would point in the same direction for the health effects as uneven exposure across a city would have the same mechanism via health. The paper needs a fuller discussion of the mechanisms at play. For other covariates the authors should use city-level measures from census statistics which would allow for additional controls.
---

VERSION 1 – AUTHOR RESPONSE

Reviewer Name: Nick Bailey

Comment: "3. The paper uses survey data on household resources and experience of mortality across a number of Indian cities to try to examine whether spatial segregation has an impact on mortality. While the topic is clearly relevant to the journal and the data appear to be of reasonable quality, I believe the design has a serious problem. I also felt that there were weaknesses in framing of the work, in the theory or motivation, and in the discussion or interpretation. All could be remedied but I think the paper cannot be published at this stage."

Response: We thank the reviewer for his careful consideration of the paper and suggestions to make the study stronger. We agree with all the suggestions he has made and have revised the paper accordingly.

Comment: "4. I think there is a basic methodological problem with the paper. The Index of Isolation is computed for districts (cities). It has a maximum of 1 (where the poor are only ever found in neighbourhoods with 100% poverty) but it has a minimum which is equal to the proportion poor in the district (city) as a whole. This occurs where the poor are dispersed in equal concentrations in all neighbourhoods. It cannot go lower than this and the authors are wrong therefore to state that it has a minimum value of zero (page 6, line 27). Cities with more poverty will therefore tend to have a higher Index value without there necessarily being any more segregation. That is why the correlation of these two at District level is so strong (0.9 – page 7, line 39).

The implication is that the correlations of district-level Index of Isolation and district-level premature mortality (Figure 1) are at least in part simply correlations of poverty (illiteracy) and mortality. Likewise, the modelling results (Table 2) which suggest a relationship between Isolation and mortality will be at least in part simply showing the impacts of poverty.

Comment: There is, however, a simple correction factor which can be applied here which removes the effect of composition, as Massey and Denton's (1988) paper noted, citing work by Bell (1954). This ought to produce a much greater separation of district poverty and segregation (although I would expect some correlation to remain), and hence permit both to be controlled in the same models."

Response: We have now calculated the Isolation index correction factor ("adjusted isolation index) as suggested and added that to the analyses (see Table 3 for the results with the adjusted Isolation index and the Results for the reporting of those results).

Comment: At present the two are modelled separately (compare models 2 and 4 in Table 2). The effects are near-identical overall and when treated as interactions with household poverty (models 3 and 5). On the latter, the paper claims there is a major difference here presumably focussing on the significance tests. The estimated regression coefficients are very similar.

Response: We now model the association of the adjusted isolation index with premature mortality alongside the district (city) level poverty rates (in the original manuscript, these were modelled separately due to collinearity issues)- see Table 3. We also present similar models where we estimate the association of the unadjusted isolation index and the index of dissimilarity with premature mortality, controlling for the district level poverty rate and all the other covariates. What is striking from the results presented in Table 3, is that the pattern of the associations is similar whether we use the unadjusted or adjusted isolation index or the index of dissimilarity. The interaction term between household poverty and the measures of segregation (adjusted/unadjusted isolation index or the dissimilarity index) suggests that poor people living in more socioeconomically segregated cities have higher risks of premature mortality than poor people living in less socioeconomically segregated cities. And this interaction between socioeconomic segregation and household poverty is independent of city (district level) poverty rates.

Response: "5. In the results summary (line 25) and in the findings (page 8, line 40), I think the paper gives an exaggerated impression of the scale of the effects observed. The scale is described in relation to a unit increase in the isolation index. As that is beyond the limits of possible change (see above), it is not an appropriate basis for reporting. The observed range is less than 0.5 (Figure 1). A better basis might be change in mortality in relation to a one standard deviation change in the Index of Isolation for the set of districts, perhaps. That would of course make the reported effect correspondingly smaller – somewhere around 2% change in mortality."

Response: For the analyses presented in Table 3, we have now converted the isolation index (and the index of dissimilarity) into z-scores, so the coefficients in the table reflect standard deviation units. The x-axis in Figure 2 now represents standard deviations of the (adjusted) isolation index. Accordingly, we have revised the "effect sizes" reported in the original manuscript to the units of SD of the isolation index.

Comment: "6. The paper risks over-generalising. Segregation is a complex, multi-dimensional concept as Massey and Denton's (1988) paper shows. In this paper, the focus is on the dimensions of 'exposure' or rather its converse, isolation. It might be better to refer to isolation throughout the analytical stage. This is made more important, perhaps, by the use of illiteracy as the measure of poverty so there is a twofold move – from the illiteracy/isolation relationship to poverty/segregation."

Response: We apologise for the lack of clarity and language in the original manuscript. We agree that the index of isolation is just one dimension of socioeconomic segregation. However, we have now also presented analyses with the index of dissimilarity (reviewer 2's suggestion) and the results are similar to the index of isolation. We have referred specifically to both indices separately in the results section, but occasionally, in the discussion section we refer to measures of socioeconomic segregation to allude to the common pattern of results arising from the index of isolation and index of dissimilarity. We hope that is acceptable. We acknowledge in the limitations that there are other dimensions of segregation that we do not measure in this study.

Comment: "7. The paper uses the term segregation too loosely in another sense. Segregation is an attribute of 'cities' or systems of neighbourhoods. Individual neighbourhoods cannot be 'segregated' – they are only poorer or richer. A poor household in a poor neighbourhood can be more isolated but not segregated. There are numerous examples where the paper uses 'segregation' as a term related to the neighbourhood."

Response: We apologise again for our lack of clarity in the original manuscript. We agree, segregation is a feature of cities, and have revised our former references to neighbourhood segregation to now refer to socioeconomic segregation of cities.

Response: "8. The first sentence of the abstract suggests that increased socio-economic segregation might not occur along with urbanisation when it is hard to see how it would not. The same point is made at start of the introduction (p4, line 3). Any segregation in rural communities would be very much more limited so urbanisation must imply a shift to more segregated contexts. In any case, the paper does not compare urban and rural but explores variations between cities so this statement is not needed."

Response: We have now amended our abstract and deleted "in some contexts" to the statement "Although urbanisation is generally associated with poverty reduction in developing countries, in some contexts, it also results in increased socioeconomic segregation of the poor." Similarly, in the introduction, we deleted "is often" from the sentence "Urbanisation in developing countries is often accompanied by an increasing spatial concentration of poverty in cities."

Comment: "9. The paper argues that spatial inequalities in health persist even after controlling for individual/household factors (page 4, line 22) and argues that this is evidence for contextual effects on health. This needs to be much more cautiously stated. These observations can never fully account for individual/household factors so cannot be taken as strong evidence for contextual effects, not least because there is such an obvious risk of selection bias: the 'poor' living in poorer neighbourhoods are likely to be poorer than those in better neighbourhoods in ways we have not observed. These selection effects are central to the neighbourhood effects literature."

Response: We completely agree that we have not been able to deal with selection factors (that we have not controlled for in the analyses) that could be behind apparent contextual effects. Accordingly, we have now revised the language of the results and discussion to take out reference to the word "effect" and replace with "associations". We also refer to potential selection bias as one of the limitations of the analyses.

Other minor points

Comment: "10. The outcome measure is a very crude mortality rate which ignores age composition."

Response: We agree that the outcome measure of under 60 mortality was very crude- this was because of the small numbers of deaths at the household level. If we had attempted more age-specific mortality analyses, this would have resulted in unstable models with zero cell counts. We now acknowledge the limitation of this crude mortality outcome measure.

Comment: "11. For the survey, no information is provided on response rates which give some indication of data quality."

Response: We now provide the survey response rates in the description of the dataset (Methods). "The overall household response rate, the number of households interviewed per targeted 100 households, was 94 percent".

Comment:

Comment: "12. Table 2 – reference category for household illiteracy is presumably 'none illiterate'."

Response: We have now changed the reference category for household illiteracy in Table 3.

Reviewer: 2

Reviewer Name: Trevon Logan

Comment: "13. The authors should also use the dissimilarity index in addition to isolation to measure the effects. Although the two measures are different they would point in the same direction for the health effects as uneven exposure across a city would have the same mechanism via health."

Response: We have now calculated the index of dissimilarity and report the results in Table 3 and on page. The pattern of results are very similar to the index of isolation (both adjusted and not adjusted for population composition)- the interaction term between household poverty and either measure of socioeconomic segregation (isolation index or dissimilarity index) showed that poor households living in more socioeconomically segregated cities were at increased risk of premature mortality than poor households living in less segregated cities. This suggests, as the reviewer suggests, that the city level mechanisms leading to increased mortality risks may be similar between the two measures of socioeconomic segregation.

Comment: "14. The paper needs a fuller discussion of the mechanisms at play."

Response: We have included in the introduction section, a discussion of potential mechanisms linking socioeconomic segregation to premature mortality risk. In the discussion section, we refer to the main causes of death in India, which include chronic and infectious diseases, as well as injuries. We also discuss how poor people living in more socioeconomically segregated cities may be more at risk of such causes of death than poor people living less segregated cities. We refer explicitly to pollution and lack of green space factors (for respiratory, cardiovascular disease and cancers), poor public transport and road infrastructure (for injuries) and the lack of water, sewage and health service factors (for infectious diseases). Levels of these contextual risk factors may be more adverse for poorer people living in more segregated cities than poorer people living in less segregated cities. In less segregated cities, the poor are likely to share some of the living conditions of the rich, reducing their exposure to these contextual risk factors for mortality. In contrast, in more socioeconomically segregated cities, where the poor live in neighbourhoods that are isolated from the rich, they are more likely to be exposed to these contextual risk factors.

Comment: "15. For other covariates the authors should use city-level measures from census statistics which would allow for additional controls."

Response: We now include the city (district level) population (per 100,000) and the percentage of city population that are scheduled caste or scheduled tribe (SCST). These were derived directly from the survey (from weighted district estimates) as the survey is meant to be representative of the surveyed districts (or in other words, cities) and the survey occurs around mid-way between censuses (resulting in potentially inaccurate city level estimates if we had used census data). The proportion of population that are SCST significantly predicted premature mortality in the expected direction. Additional covariates were considered but we could not include many more district level covariates in the model as the district level sample size was 59, and our main independent variables were the segregation measures at the district level. So to summarise, in our regression models, we control for district level poverty rates, district level population and district level SCST % (and also a number of household level factors), when estimating the association between district level measures of segregation and premature mortality.